# Investigating Markers of the NLRP3 Inflammasome Pathway in Alzheimer’s Disease: A Human Post-Mortem Study

**DOI:** 10.3390/genes12111753

**Published:** 2021-10-30

**Authors:** Hao Tang, Michael Harte

**Affiliations:** 1Division of Pharmacy and Optometry, School of Health Sciences, Faculty of Biology, Medicine and Health, University of Manchester, Manchester M13 9PL, UK; hao.tang@manchester.ac.uk; 2Department of Neurology, The First People’s Hospital of Yunnan Province, Kunming 650032, China

**Keywords:** NLRP3 inflammasome, neuroinflammation, Alzheimer’s disease, gene expression, post-mortem

## Abstract

Neuroinflammatory mechanisms with glial cell activation have been implicated in the pathogenic process of Alzheimer’s disease (AD). Activation of the NLRP3 inflammasome is an essential component of the neuroinflammatory response. A role for NLRP3 activation in AD is supported by both in vitro and in vivo preclinical studies with little direct investigation of AD brain tissue. RNA expression of genes of three glial cell markers, HLA-DRA, AIF-1 and GFAP; the components of the NLRP3 inflammasome NLRP3, ASC, and caspase-1; and downstream pre-inflammatory cytokines IL-1 β and IL-18, were investigated in the temporal cortex of AD patients and age- and sex-matched controls. Protein expression of GFAP was also assessed. Increases in both mRNA and protein expression were observed for GFAP in AD. There were no significant changes in other NLRP3 activation markers between groups. Our results indicate the involvement of astrocyte activation in AD, particularly in more severe patients. We found no evidence for the specific involvement of the NLRP3 inflammasome.

## 1. Introduction

Alzheimer’s disease (AD)—the most common form of dementia—is a major cause of disability and premature death among older people worldwide [1,2]. Impaired cognitive function, behavioral changes, and consequent lack of independence in AD patients, along with an increasing prevalence of AD, substantially challenges the public health system. Several hypotheses have been postulated to explain the pathogenesis of AD. The pathology of AD is characterized by the appearance of two hallmarks: senile plaques (SP), formed by progressive pathological assemblies of the amyloid beta (Aβ) peptide and neurofibrillary tangles (NFTs), which are comprised of hyperphosphorylated microtubule-associated protein tau [3]. These pathological markers may be indicative of pathogenic processes contributing to neuronal damage and death, although no single theory is sufficient in explaining the cause or causes of AD. 

Neuroinflammation has been proposed as a factor involved in the pathogenesis of AD [4]. Neuroinflammation refers to the inflammatory response in the central nervous system (CNS), of which microglia and astrocytes are the prominent cellular mediators [5]. It may be initiated acutely in response to cellular damage or the invasion of pathogenic microorganisms in the CNS, enabling the maintenance of neural homeostasis [5]. Nevertheless, the persistent activation of microglia and/or astrocytes can also lead to deleterious inflammatory reactions during chronic stimulation by infections or in response to uncontrolled immune triggers, such as Aβ and tau oligomers [6,7]. The inflammatory cascade may then drive further amyloidogenesis and tauopathy, accelerating the disruption of neuronal function [8,9,10,11].

One potentially important component of this neuroinflammatory process is activation of the NOD-like receptor pyrin domain containing 3 (NLRP3) inflammasome. Functioning as sensors for multiple stimuli, inflammasomes are the determinant regulators of the innate immune reaction, resulting in the secretion of proinflammatory cytokines. Different inflammasomes respond to particular pathogenic agents. NLRP3 can be activated by a wide range of aggregated substances, including the proteinaceous deposits associated with several neurodegenerative disorders, by detecting the damage to internal membranes [12]. The NLRP3 inflammasome is composed of three proteins: a cytosolic sensor molecule NLRP3; an apoptosis-associated speck-like protein containing a caspase-activating recruitment domain (ASC, coded by *PYCARD* gene); and an effector molecule, cysteine protease pro-caspase-1 (CASP1) [13]. By assembling into a large speck-like structure, ASC provides the platform together with NLRP3 for activating pro-caspase-1 [14]. The subsequent production of active caspase-1 is responsible for the cleavage of pro-interleukin-1β (pro-IL-1β) and pro- interleukin-18 (pro-IL-18) into their mature forms, which then trigger inflammatory responses by stimulating downstream cytokines and chemokine receptors [15]. A potentially important role for NLRP3 inflammasome activation is indicated in AD by emerging pre-clinical animal model and cell model studies [16]. These have shown that the acutely activated NLRP3 inflammasome can exacerbate amyloid and tau pathologies [17,18,19], while both the accumulation of Aβ and aggregation of tau have been found to result in NLRP3 inflammasome activation [20,21]. However, there is little supporting data from human studies, other than controversial findings indicating unchanged mRNA and protein expression of NLRP3 in small studies of post-mortem AD brain tissue [22,23].

The aim of this study is to investigate evidence for NLRP3 inflammasome activation in AD using brain tissue collected at post mortem from moderately and severely affected AD subjects and neuropathologically unaffected control subjects. The brain region examined in this project is the temporal cortex, as it is particularly affected by underlying molecular pathologies in AD [24]. We determined relative gene expression of three glial cell markers: Major Histocompatibility Complex, Class II; DR Alpha (HLA-DRA); Allograft Inflammatory Factor 1 (AIF-1) and Glial Fibrillary Acidic Protein (GFAP); the components of the NLRP3 inflammasome, NLRP3, ASC, and caspase-1; and downstream pre-inflammatory cytokines IL-1 β and IL-18. We also assessed GFAP and caspase-1 protein expression.

## 2. Results

### 2.1. RNA and Protein Expression of GFAP Was Increased in Advanced AD

Gene expression of *GFAP* as a marker for astrocyte activation was significantly associated with subject groups. It was significantly elevated in advanced AD but not significantly so in moderate AD (Figure 1). 

Determination of GFAP protein showed this to be significantly increased above control values in AD (Figure 2b) but not reaching significance in individual AD subgroups (Figure 2c).

### 2.2. RNA Expression of AIF1 and HLA-DRA Was Unchanged in AD

Gene expression of indicators of microglial activation, *AIF1* (Figure 3a) and *HLA-DRA* (Figure 3b), was not significantly associated with subject groups. 

### 2.3. RNA and Protein Expression of NLRP3 Inflammasome Was Unchanged in AD

Relative gene expression of the components of the NLRP3 inflammasome, *IL1B* and *IL18* is shown in Figure 4. No significant change was observed in any of these mRNAs in AD or either AD subgroup. 

Due to the low sensitivity, only pro-caspase-1 protein was detected by capillary electrophoresis immunoblotting. The protein level of pro-caspase-1 was unchanged between groups (Figure 5).

Both in the whole sample and within the AD group, no significant association of any gene expression measure with apolipoprotein E genotype or sex was apparent, except for IL1beta, which showed an increased expression in males (*t*(113) = −2.21, *p =* 0.03).

Re-analysis of gene expression data including RIN as a covariate made essentially no difference to the results, with no significant findings from the NLRP3 components and consistently elevated expression of GFAP, other than the small increase in *HLA-DRA* expression in advanced AD showing significance (GLM-UniANOVA, *F* = 3.717, *R*^2^ = 0.133, *p =* 0.028). 

## 3. Discussion

Although the pathological process in AD remains ill-defined, the evidence that inflammation plays a role is strong. The current findings demonstrating an increase above control values in GFAP mRNA and protein expression, particularly in subjects with advanced AD pathology, support this. Such increases in GFAP are fairly well established in the brain in AD and are consistent with an activation of astrocytes, an indication of inflammatory response [25,26,27,28,29]. Interestingly, recent studies all observed the strong correlation between plasma GFAP and AD pathology, suggesting its role as an early marker of AD [30,31,32]. 

The evidence for changes in microglia in the AD brain in this study was equivocal. Two markers of microglia have been chosen for study based on the previous AD human post-mortem research [33]. We found no increase in mRNA expression of AIF1, also known as ionized calcium-binding adapter molecule 1 (Iba1). This is a marker for both activated and resting microglia. A review of previous studies found that the change of AIF-1 expression is not consistently reported in AD tissue [33], with many reports finding no change, although very few studies involved mRNA analysis [34]. A more recent study of protein immunoreactivity in the temporal cortex in AD also found no increase in AIF-1, again in the presence of an elevation of GFAP [35]. The other microglial marker we studied was *HLA-DRA* mRNA, coding for a component of the major histocompatibility complex II. This is considered to have more specificity for activated microglia than AIF-1 and has been reported as generally elevated in AD and correlated with AD plaque stage and clinical dementia rating [33], although only one previous mRNA study, in hippocampal tissue, was identified. We did not observe an unequivocal change in this marker, although the small increase that was found did emerge as significant after correction for RNA integrity. 

A mechanism implicated in the inflammation associated with several chronic disease processes is activation of the NLRP3 inflammasome. This has substantial evidence supporting its involvement in AD, primarily from in vitro and animal model paradigms, as described in the introduction [17,18,19]. The NLRP3 inflammasome in the brain is primarily associated with microglia [36,37]; the evidence from this post-mortem study of a small change in *HLA-DRA* mRNA expression provides some support for activation of microglia in AD, albeit to a much lesser extent than for the astrocyte marker GFAP. Direct measures of mRNA for proteins comprising the NLRP3 inflammasome do not, however, provide any support for its activation in AD. We did not detect significant differences in other NLRP3 inflammasome activation markers, including *NLRP3*, *PYCARD* and *CASP1*, as well as the downstream pro-inflammatory cytokines *IL1B* and *IL18*, between AD and control. This is nevertheless consistent with other post-mortem studies that reported no change in mRNA and protein expression of NLRP3, as well as mRNA expression of *PYCARD* and *CASP1* in AD [22,23,38] (Table 1). Heneka and colleagues found an elevation of caspase-1 protein in the AD brain, while the other two protein components of the NLRP3 inflammasome were not reported [17]. Based on this, caspase-1 protein expression was investigated in our cohort. Due to the low sensitivity to cleaved/activated caspase-1, only pro-caspase-1 was detected, which showed no significant changes between groups, consistent with the mRNA expression (Figure 5). While another microarray study found no differences in *IL1B* expression in the frontal and occipital regions between AD and control [39] (Table 1), our investigation has a much larger sample size with both clinical and neurological confirmation. 

The lack of concordance with findings from model systems is a concern; however, the profound differences in the time course of development of AD pathology in humans (years) compared to cellular (days) and animal (weeks-months) models may differentially influence the involvement of inflammatory processes. The gene expression profile of microglial activation in AD patients was distinct from existing mouse models [40], while glia can adopt a reactive nonphysiological phenotype in vitro, changing their gene expression features [41,42]. Thus, developing animal models that recapitulate multiple facets of AD and novel three-dimensional cell culture models are proposed to help us further understand the role of inflammation in the progression of AD [4,43].

There are inevitably many limitations to this post-mortem study. While we have a relatively substantial sample size in comparison to many such human brain studies, our results are limited to the one brain region investigated. It would be important in future studies to determine whether the astrocytic activation indicated by elevated GFAP mRNA is found in brain regions affected by AD other than the temporal cortex, particularly the hippocampus, and whether the absence of evidence for NLRP3 activation also extends to these other areas. Changes in the cellular expression of markers may not always be reflected by results in tissue due to recruitment or depletion of the relevant cell [44]. While the determination of mRNA can provide a direct measure of gene transcription, it may not always reflect the levels of, or changes in, translated protein end-product [45]. Furthermore, the use of housekeeping genes to control for variability in mRNA yield and stability may be far from ideal, particularly in post-mortem human tissue that may be subject to a variety of factors, such as agonal state, postmortem interval, and brain pH, that influence and may severely compromise mRNA integrity [46,47]. Nevertheless, the use of samples from well-matched non-AD subjects provides some control for many of these confounding effects.

## 4. Materials and Methods

### 4.1. Source of Human Brain Tissue and Research Ethics Committee Approval 

Post-mortem brain tissue samples were obtained from the Manchester Brain Bank at Salford Royal NHS Foundation Trust (University of Manchester). The Manchester Brain Bank has been approved by the Newcastle & North Tyneside 1 Research Ethics Committee (REC reference 09/H0906/52+5). 

On the basis of the reported tau and Abeta amyloid pathology in each brain, samples of temporal cortical tissue from 37 control subjects (Braak Stage 0-II) and 39 each of moderate (Braak Stage III-IV) and advanced AD (Braak Stage V-VI) subjects were studied. Demographic data and apolipoprotein E genotype for each group are listed in Table 2. Group differences in age were observed, and this variable was included as a covariate in gene expression analysis.

Standard procedures were carried out, including post-mortem brain sample storage, handling, and examination according to UK research ethics. Unfixed samples were stored at −80 °C immediately after dissection until tissue homogenization. Research was conducted in compliance with the principles of the Declaration of Helsinki. 

### 4.2. Quantitative Real-Time PCR 

RNA expression was quantified using reverse transcriptase semi-quantitative real-time PCR (RT-qPCR). All the procedures were performed according to the MIQE guidelines [48]. The MIQE checklist for this study is in Appendix A.

#### 4.2.1. RNA Isolation and cDNA Synthesis

Total RNA extraction was performed using a TRIzol Reagent (Thermo Fisher, UK), followed by RNA clean up using a Monarch RNA Cleanup Kit (New England Biolabs, UK), according to the manufacturer’s instructions. RNA purity and concentrations were assessed using a NanoDrop 2100 (Thermo Fisher, UK). RNA concentrations ranged from 184.8 to 593.5 ng/µL. An optical density at wavelength 260/280 nm range of 2.01–2.11 was chosen for this study. RNA integrity number (RIN) was assessed by an Agilent 2200 TapeStation System. DNA contamination was assessed in RT-qPCR by using RNA without reverse transcription step—cDNA synthesis. Single-stranded cDNA was synthesized from 1 ng of total RNA using M-MLV Reverse Transcriptase (Thermo Fisher, UK). The cDNA concentrations were measured on a NanoDrop 2100 and diluted to 500 ng/µL in RNase-free water.

#### 4.2.2. Selection of Candidate Reference Genes and PCR Primer Design

Six reference genes (RG) were selected for our cohort (Appendix A) according to previous studies (Appendix A). The stability of RGs was ranked by RefFinder [49], using the raw threshold cycle (Ct) value of RGs. All samples were run in triplicates, and the top two stable RGs—RPL13A and GAPDH—were chosen for delta Ct analysis (Appendix A). 

Primer pairs were designed by using Primer-BLAST (NCBI, http://www.ncbi.nlm.nih.gov/tools/primer-blast/, accessed on 11 February 2019, and exon junctions were included wherever possible to avoid amplification of genomic DNA. Primer pairs with the least probabilities of amplifying nonspecific products as predicted by the NCBI PrimerBLAST were selected. Primers were synthesized by ThermoFisher. For each primer pair (Appendix A), the optimal annealing temperature (Ta) was calculated using an online Tm Calculator (ThermoFisher) based on the modified Allawi & SantaLucia’s thermodynamics method [50]. The amplification efficiency (E) and coefficient of correlation (R2) were determined for each primer pair by at least 4 points on the standard curve. Primer specificity was analysed by dissociation curve (Appendix A).

#### 4.2.3. Reverse Transcriptase Semi-Quantitative Real-time PCR (RT-qPCR)

RT-qPCR reactions were carried out using Power SYBR Green PCR Master Mix (Thermo Fisher, UK) on a 7900HT Real-Time PCR System with a 384-well format (Applied Biosystems, UK). The final volume for each reaction was 10 µL, with 1 μL (0.25 μM) of corresponding primers (Appendix A), 1 µL (500 ng/µL) of total cDNA, 3 µL of RNase-free water, and 5 µL of SYBR green. A positive control (cDNA sample of control group which showed high Ct value of reference genes and a negative water control) was included for each primer pair on each plate. All samples were run in triplicates. The thermal cycler parameters were as follows: UDG activation at 50 °C for 2 minutes and DNA polymerase activation at 95 °C for 10 minutes, followed by amplification of cDNA for 40 cycles with denaturation at 95 °C for 15 seconds and annealing/extension at respective annealing temperature for each gene (Appendix A) for 1 minute. Dissociation curve analyses were carried out at the end of each run for PCR product verification (Appendix A). The intra-assay variation was controlled as standard deviation of triplicates <0.2. The inter-assay variation was analysed by using raw the Ct value of RGs, and there was no significant variation between plates (Appendix A).

### 4.3. Protein Analyses: Capillary Electrophoresis Immunoblotting

Protein expression of GFAP and caspase-1 was quantified using automated capillary electrophoresis immunoblotting.

#### 4.3.1. Protein Extraction and Purification

A measurement of 40–60 mg of brain tissue was homogenized by a 10-fold volume (mg:μL) of lysis buffer (pH 7.4, 10 mM Trizma base, 2 mM EDTA, 320 μM sucrose (Sigma-Aldrich, UK)). A cOmplete Protease Inhibitor Cocktail (Sigma-Aldrich, UK), phenylmethylsulfonyl fluoride (PMSF), and sodium orthovanadate were added to the lysis buffer immediately before use at 4%, 1%, and 1% (*v*/*v*) concentrations, respectively. The differential centrifugation method was used for protein purification [51].

#### 4.3.2. Protein Quantification

Automated capillary electrophoresis immuno-quantification was conducted on a Wes instrument for target protein analyses, according to the manufacturer’s protocol (ProteinSimple) [52]. In brief, protein concentration was assessed using the Bradford assay, then optimization of sample concentration and primary antibodies was carried out for each protein. The sample was diluted to the optimized concentration—0.4 mg/mL. After being mixed with a master mix containing dithiothreitol and a fluorescent molecular weight marker in a ratio of 5:1, samples were heated at 95 °C for 5 minutes for denaturation. Then, prepared samples, blocking reagent (Wes antibody diluent), target protein primary antibodies, horseradish peroxidase (HRP)-conjugated secondary antibodies, and chemiluminescent substrate were loaded into the allocated wells of a Wes plate, which pre-loaded with sample and stacking matrices. Instrument default settings were used. The size-based separation electrophoresis, immobilization, and immunodetection were automatically processed in the capillary array system, and the produced chemiluminescence was detected at multiple exposure times and quantified by the ProteinSimple v3.1 software. The primary antibodies used were rabbit anti-GFAP (1:200, ab68428, Abcam, UK) and recombinant anti-Caspase-1 (1:200, ab207802, Abcam, UK). The total protein value detected by Total Protein assay (ProteinSimple) of each sample was used for normalization. 

### 4.4. Statistical Analysis

mRNA analysis was carried out using the 2(-Delta Delta Ct) Method [53]. Raw Ct values were normalized using the geometric mean of two RGs as an endogenous internal standard. For our cohort, the most stable RGs were RPL13A and GAPDH (Appendix A). Data are displayed as mean ± SD or mean ± SEM. Gaussian distribution was evaluated using the Shapiro-Wilk normality test. Homogeneity of variance was evaluated using Levene’s test. Statistical analyses were conducted in IBM SPSS Statistics 23 and GraphPad Prism 7 software using non-paired parametric Student’s t-test, Welch’s t-test for unequal variances, one-way ANOVA followed bya post hoc test, and Pearson correlation or Spearman rank correlation. Statistical significance was set at *p*-value ≤ 0.05 (**p* ≤ 0.05, ** *p* ≤ 0.01, *** *p* ≤ 0.001). 

## 5. Conclusions 

In conclusion, we find that while there is evidence for a substantial activation of astrocytes in the temporal cortex in late-stage AD, there is no apparent activation of the NLRP3 inflammasome, contrary to the findings in relatively acute model systems mimicking AD pathogenesis.

## Figures and Tables

**Figure 1 genes-12-01753-f001:**
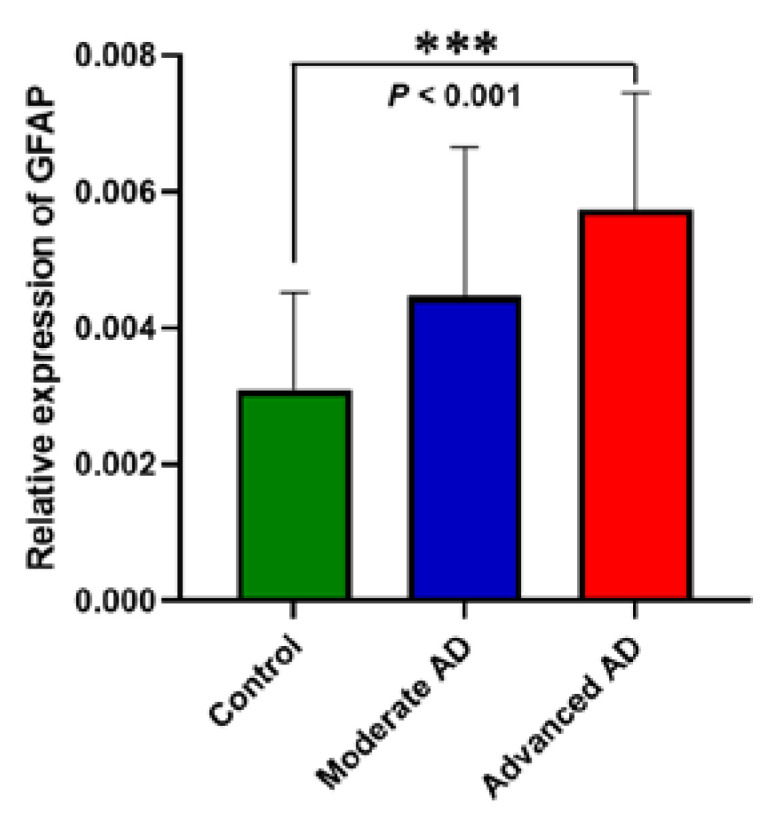
RNA expression of GFAP. Quantitative PCR analysis of *GFAP* expression relative to geometric mean of *GAPDH* and *RPL13A* housekeeping genes expression. Significant elevation of *GFAP* expression was observed in advanced AD (*p* < 0.001, 95% confidence interval= (0.85, 3.40)) but not in moderate AD (*p =* 0.557, 95% confidence interval = (−0.58, 1.97)). Control, *n =* 37; moderate AD, *n =* 39; advanced AD, *n =* 39. Data are presented as the mean ± SEM and analyzed by the one-way ANOVA test followed by the Dunnett post hoc test. *** *p* ≤ 0.001.

**Figure 2 genes-12-01753-f002:**
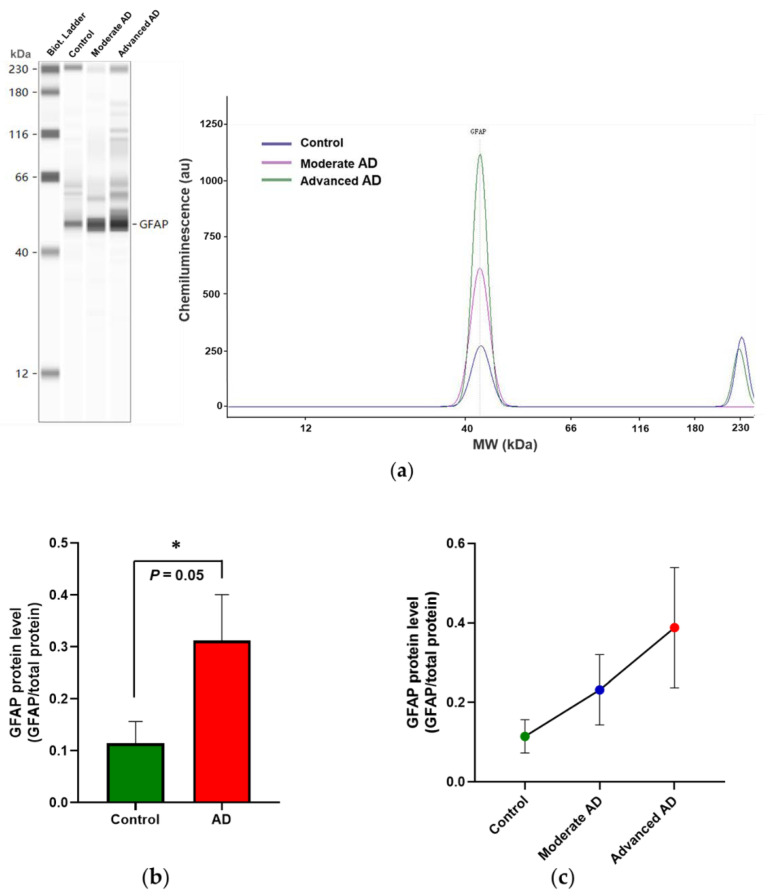
Automated capillary electrophoresis immuno-quantification analysis of GFAP protein expression. (**a**) Results are shown as gel-like image views in left panel and electropherograms in the right panel, showing decreasing intensities of bands and decreasing peak areas in AD subgroups. (**b**) GFAP protein expression significantly increased in AD (M = 0.31, SD = 0.49) compared with control (M = 0.11, SD = 0.17) (*t*(112) = −2.02, *p* = 0.05). (**c**) Further analysis of GFAP protein expression in individual AD subgroups (one-way ANOVA, *F*(2, 112) = 1.83, *p* = 0.17). Control, *n* = 37; moderate AD, *n* = 39; advanced AD, *n* = 39. Data are presented as the mean ± SD. * *p* = 0.05.

**Figure 3 genes-12-01753-f003:**
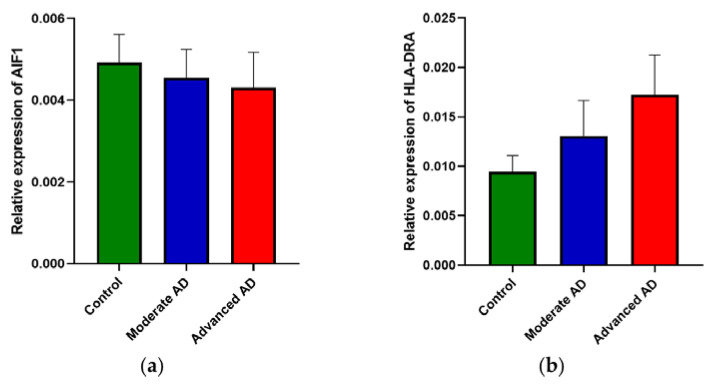
RNA expression of *AIF1* and *HLA-DRA*. There was no significant difference between subject groups of (**a**) *AIF1* expression (one-way ANOVA, *F*(2, 112) = 1.98, *p* = 0.143) and (**b**) *HLA-DRA* expression (one-way ANOVA, *F*(2, 112) = 0.62, *p* = 0.54). Control, *n* = 37; moderate AD, *n* = 39; advanced AD, *n* = 39. Data are presented as the mean ± SEM.

**Figure 4 genes-12-01753-f004:**
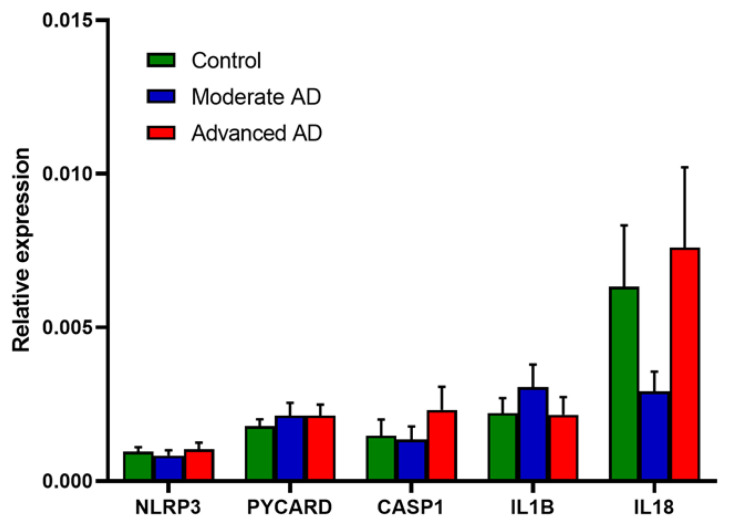
RNA expression of the components of the NLRP3 inflammasome and downstream pro-inflammatory cytokines. Control, *n* = 37; moderate AD, *n* = 39; advanced AD, *n* = 39. Data are presented as the mean ± SEM.

**Figure 5 genes-12-01753-f005:**
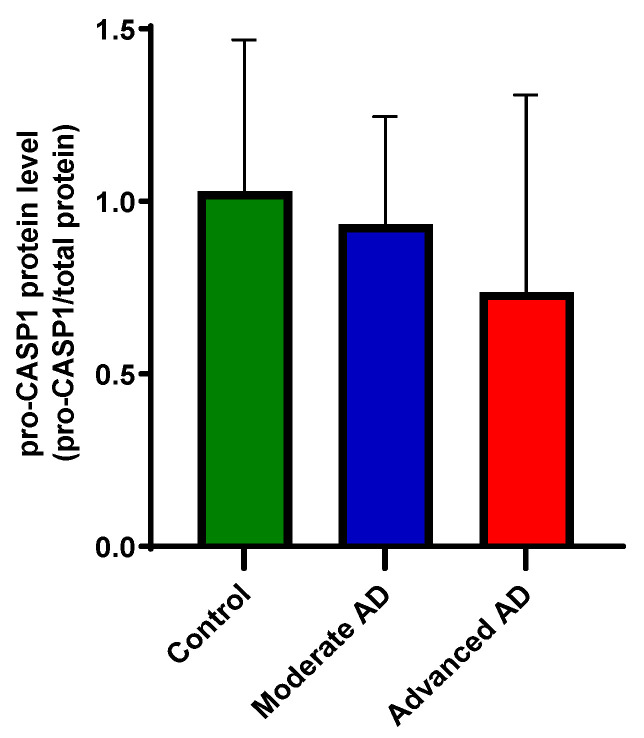
Protein expression of caspase-1. The detected pro-caspase-1 level was unchanged between groups (one-way ANOVA, *F*(2, 50) = 1.91, *p =* 1.58). Control, *n =* 17; moderate AD, *n =* 17; advanced AD, *n =* 19. Data are presented as the mean ± SD.

**Table 1 genes-12-01753-t001:** Gene and protein expression of the NLRP3 inflammasome in human post-mortem studies of Alzheimer’s disease.

Study	Sample Size	Brain Region	Technique	Markers of NLRP3 Inflammasome		
				NLRP3	*PYCARD*/ASC	*CASP1*/Caspase-1
Heneka et al. [17]	AD, *n* = 12 NC, *n =* 8	FC HC	WB	NR	NR	↑ in AD
Cribbs et all. [22]	AD, *n =* 26 NC, *n =* 33	EC	Microarray			↑ in NC
		HC				
		SFG				
		PCG				
		SFG	qPCR	no change	no change	
Liu et al. [23]	AD, *n =* 5 NC, *n =* 5	FC FC	qPCR WB	no change no change	↑ in AD ↑ in AD	NR
Li et al. [38]	AD, *n =* 6–7 NC, *n =* 6–7	FC	WB	no change	↑ in AD	↑ in AD
Present study	AD, *n =* 78 NC, *n =* 37	TC	qPCR	no change	no change	no change

Abbreviations: AD, Alzheimer’s disease; NC, non-demented control; FC, frontal cortex; EC, entorhinal cortex, HC, hippocampus; SFG, superior frontal gyrus; PCG, post-central gyrus; TC, temporal cortex; NR, not reported; WB, Western blot; qPCR, quantitative real-time PCR. ↑ denotes increased expression.

**Table 2 genes-12-01753-t002:** Demographic and clinical characteristics of study subjects.

	Age of Death (Years)	Post-Mortem Interval (Hours)	APOE Genotype	
			ε4 Non-Carrier	ε4 Carrier
Control group				
All (*n =* 37)	82.6 ± 11.5	81.2 ± 36.9	32	5
Male (*n =* 19)	84.5 ± 9.4	85.7 ± 41.4	17	2
Female (*n =* 18)	80.7 ± 13.3	76.5 ± 31.8	15	3
Moderate AD group				
All (*n =* 39)	85.5 ± 6.3	79.4 ± 38.4	21	18
Male (*n =* 20)	84.0 ± 6.5	81.2 ± 37.0	9	11
Female (*n =* 19)	87.1 ± 6.0	77.5 ± 40.6	12	7
Advanced AD group				
All (*n =* 39)	78.1 ± 8.8 *	86.2 ± 43.0	10	29
Male (*n =* 20)	73.3 ± 4.7 *	82.7 ± 48.9	4	16
Female (*n =* 19)	83.0 ± 9.3	89.6 ± 37.1	6	13

* The age of death of the advanced AD group was younger than other two groups, but it was not correlated with any of the quantification values of mRNA or protein expression assessed in this study.

## Data Availability

The expression data presented in this study are available on request from the corresponding author. Restrictions apply to the availability of clinical data which was obtained from the Manchester Brain Bank.

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
