# Peer review of "Investigating Markers of the NLRP3 Inflammasome Pathway in Alzheimer’s Disease: A Human Post-Mortem Study"

_genes, 2021, doi:10.3390/genes12111753_

Round 1

Reviewer 1 Report

Manuscript ID: genes-1416956

Title: Investigating markers of the NLRP3 inflammasome pathway in Alzheimer’s disease: A human post-mortem study.

Authors: Hao Tang, Michael Harte *

Rating the manuscript:

In this article, the authors have investigated the markers of NLRP3 inflammasome pathway and glial cell activation pathway in Alzheimer’s Disease. Neuroinflammatory mechanisms with glial cell activation have been implicated in the  pathogenic process of Alzheimer’s disease (AD). Activation of the NLRP3 inflammasome is an essential component of the neuroinflammatory response. A role for NLRP3 activation in AD is supported by both in vitro and in vivo preclinical studies with little direct investigation of AD brain tissue. In this manuscript the authors investigated RNA expression of three glial cell markers: HLA-DRA, AIF-1 and GFAP, the components of the NLRP3 inflammasome: NLRP3, ASC and Caspase-1, and downstream pre-inflammatory cyto-15 kines: IL-1 β and IL-18, in temporal cortex of AD patients and age and sex matched controls. Protein expression of GFAP was also assessed. Increases in both mRNA and protein expression were observed for GFAP in AD. There were no significant changes in other NLRP3 activation markers between groups. Their results indicate the involvement of astrocyte activation in AD, particularly in more severe patients. They did not find any evidence for the specific involvement of the NLRP3 inflammasome.  

This article is within the aims and scope of genes as it investigates genes related to astrocyte activation pathway and inflammasome activation pathway in severe AD patients.

Review Report:

There are few points that need to be addressed in the paper:

In this manuscript only glial activation marker GFAP levels changes both at the RNA and protein levels. However microglial activation markers like AIF1 and HLA-DRA and components of the NLRP3 inflammasome and downstream pro-inflammatory cytokines like NLRP3, PYCARD, CASP1, ILIB and ILI8 do not show any changes in the RNA levels by qPCR.

I would like to know if there are any changes in the protein levels. Sometimes, there are a lot of changes in the gene expression profiling regulated by small RNAs like microRNAs in AD patients. These microRNAs regulate gene expression post-transcriptionally. Thereby there might be in some cases no changes in the mRNA levels but only changes in the protein levels. So a western blot  or other quantitative protein levels analysis should be done to decipher the above.

Reviewer 2 Report

The present manuscript by Tang and Harte is well written and clear. The results are however sparse. The suggested astrogliosis is interesting and could be followed up. Additional measurements would provide important information regarding the state of the astrocytes as previously done in Creutzfeldt Jakob disease brain and other diseases. Histological analysis on the samples would also reinforce the results suggested by the qPCR.

In individual reactive microglia IBA1 expression level can be modified but the level may appears similar at the tissue level due to the active recruitment or depletion of the cells in this region. Did the author consider including CD68, a common microgliosis marker, in their analysis? If it was deliberately not chosen for the study this should be discussed. Histological evaluation using IBA1 staining would also provide important quantitative and qualitative information regarding microglia state.

Mean ± SD would be preferred over mean ± SEM. This is indicated in the figure one and should be in the legend of the other figures. The number of samples used for the different graph could be added to the figures legend to increase readability.

Suppl Fig2: “Anaylsis of inter-assay variation” typo should read Analysis

L42: provide a reference regarding pathogenic origin to AD.

Fig1, 3: transcript names should be italicized.

Round 2

Reviewer 2 Report

Dear authors, the manuscripts reads well and you have answered all my previous comments. Based on your feedback I have an additional question: why deciding to use qPCR rather than RNAseq? 

You mentioned the limited quantity of material available in your response, RNAseq would have provided more data while requiring the same quantity or less material.

This manuscript is a resubmission of an earlier submission. The following is a list of the peer review reports and author responses from that submission.